# Genetic Susceptibility to Acute Kidney Injury

**DOI:** 10.3390/jcm10143039

**Published:** 2021-07-08

**Authors:** Christian Ortega-Loubon, Pedro Martínez-Paz, Emilio García-Morán, Álvaro Tamayo-Velasco, Francisco J. López-Hernández, Pablo Jorge-Monjas, Eduardo Tamayo

**Affiliations:** 1BioCritic. Group for Biomedical Research in Critical Care Medicine, University of Valladolid, 47003 Valladolid, Spain; cjortegal@clinic.cat (C.O.-L.); egarmo@egarmo.com (E.G.-M.); alvarotv1993@gmail.com (Á.T.-V.); flopezher@usal.es (F.J.L.-H.); eduardo.tamayo@uva.es (E.T.); 2Department of Cardiovascular Surgery, Hospital Clinic of Barcelona, 08036 Barcelona, Spain; 3Department of Surgery, Faculty of Medicine, University of Valladolid, 47003 Valladolid, Spain; 4Department of Cardiology, Clinical University Hospital of Valladolid, 47003 Valladolid, Spain; 5Department of Hematology and Hemotherapy, Clinical University Hospital of Valladolid, 47003 Valladolid, Spain; 6Institute of Biomedical Research of Salamnca (IBSAL), University Hospital of Salamanca, 37007 Salamanca, Spain; 7Group of Translational Research on Renal and Cardiovascular Diseases (TRECARD), Departmental Building Campus Miguel de Unamuno, 37007 Salamanca, Spain; 8Department of Anesthesiology and Critical Care, Clinical University Hospital of Valladolid, Ramón y Cajal Ave, 47003 Valladolid, Spain

**Keywords:** acute kidney injury, predisposition, gene polymorphism, genetic variation, intensive care unit

## Abstract

Acute kidney injury (AKI) is a widely held concern related to a substantial burden of morbidity, mortality and expenditure in the healthcare system. AKI is not a simple illness but a complex conglomeration of syndromes that often occurs as part of other syndromes in its wide clinical spectrum of the disease. Genetic factors have been suggested as potentially responsible for its susceptibility and severity. As there is no current cure nor an effective treatment other than generally accepted supportive measures and renal replacement therapy, updated knowledge of the genetic implications may serve as a strategic tactic to counteract its dire consequences. Further understanding of the genetics that predispose AKI may shed light on novel approaches for the prevention and treatment of this condition. This review attempts to address the role of key genes in the appearance and development of AKI, providing not only a comprehensive update of the intertwined process involved but also identifying specific markers that could serve as precise targets for further AKI therapies.

## 1. Introduction

Disparities in kidney disease predisposition among individuals have been sharply identified [1]. The United States Renal Data System Report shows a constant increment in the occurrence of end-stage kidney disease (ESKD) among Africans that is 3.5–5-fold that of Europeans [2]. While the risk of Africans developing ESKD is 8% approximately, that of Europeans is nearly 2–3% [3].

### 1.1. Acute Kidney Injury

Acute kidney injury (AKI) is a widely held concern in hospitalized patients [4]. It entails elevated morbidity and mortality [5,6], reaching up to 50–70% [7], as there are no effective therapies to counteract, ameliorate or speed up the recovery from this condition, other than mainly supportive management [8,9,10,11,12,13,14]. Nearly 40% of patients in intensive care units (ICUs) develop AKI, as well as other concomitant states such as sepsis or hemodynamic failure [15,16]. AKI is presented by a sharp and unexpected slump in renal function, often related to other medical conditions, such as septic shock, or procedures, for instance cardiac surgery [17]. It is usually defined as a sudden increment in the serum creatinine, a deterioration in renal output or both, occurring within 48 h or less [18,19]. Nevertheless, why certain patients are more vulnerable to develop AKI than others remains unknown [20]. Thus, attempts to promptly identify individuals at risk of getting AKI are of paramount importance, but a reliable prediction of developing AKI remains a challenge.

Despite progress in the comprehension of AKI pathophysiology, its pathogenesis remains complex and incompletely understood [21,22,23], including numerous pathophysiological mechanisms that lead to sudden tubular necrosis/apoptosis and kidney dysfunction [20,24,25,26]. These comprise ischemia/reperfusion injury, complement activation, adenosine triphosphate (ATP) reduction, infiltration of leukocytes, generation of endotoxins, oxygen radicals, proinflammatory mediator generation, endothelial damage, fibrosis, microcirculatory dysfunction and, finally, cell death [24,27,28,29].

Recently, inflammatory response has been recognized as a crucial factor in AKI pathogenesis [25,30,31,32,33,34]. Infiltration of inflammatory cells has been identified in the injured kidney, which causes renal vessel damage and posterior progress to AKI [35]. These cells do not only start but also maintain the kidney damage by producing oxygen radical species, vasoconstrictors such as endothelin and impeding the release of nitric oxide, directly causing endothelial injury [30]. What is more, patients’ individual host repair and regeneration biology could also have a major influence on the etiology of AKI [36].

Genetic factors have been proposed as potentially responsible for the susceptibility and severity of AKI, explaining why only particular patients are more prone to AKI and why different patients respond to treatment distinctly [37,38,39,40,41]. This fact has led to a new form of exercise medicine, known as personalized medicine, the aim of which is for patients to be treated as individuals and not as groups [42]. A deeper understanding of how gene interaction drives health or disease might lead to a more personalized medical strategy, allowing the creation of a unique individual genetic print on which medical decisions should be based. Numerous articles have explored the relationship between different polymorphisms and AKI predisposition in various clinical scenarios [43,44]. Table 1 summarizes the evidence related to AKI susceptibility, including the study design, clinical settings, participant/population characteristics, studied polymorphisms and outcomes.

### 1.2. Controversy among Polymorphism Association Articles

Polymorphism association studies usually consist of case–control and observational studies, comparing the occurrence of a genetic variation in some individuals with a certain illness to the occurrence in a control healthy population. Furthermore, although several polymorphisms have been identified, most studies often consist of relatively small homogeneous sample size populations, which significantly restricts the ability to draw conclusions from the results of the general population and frequently gives conflicting results with mainly nonsignificant findings. In general, the results are irregular and usually variable throughout the different analyses [36]. The deficiency of a reliable and consistent association is a serious limitation given the complicated, entangled pathogenesis of the disease. This basic understanding has barely contributed to creating a holistic view of the more susceptible phenotype prone to developing AKI. What is more, some diseases are polygenic in nature, and the interactions among relevant proposed genes have not been identified. Indeed, a far more complete overview could be the application of linking analyses, in which genetically related polymorphisms are studied and where the polymorphism of a particular gene could be the promoter of another illness-inducing variation [45].

Genome-wide association study (GWAS) has come to revolutionize the identification of novel disease susceptibility genes, leading to the discovery of novel biological mechanisms and offering insight into the ethnic variation of complex traits, among others. In spite of the fact that GWAS is a powerful method to determine genotype–phenotype association, this method presents limitations such as that it can only explain a small fraction of the heritability of complex traits and does not necessarily pinpoint causal variants and genes. Moreover, it was necessary to adopt a high level of significance to account for multiple tests. This is because false-positive relationships are innumerable in genetic association reports to identify frequent polymorphisms [46]. Moreover, the low frequency of some studied SNPs affects the power of these studies [47]. Thus, a considerably large sample size is needed to achieve the optimal statistical power and minimize the spurious associations [48,49].

Association of genetic polymorphisms and AKI risk usually differs among ethnicities, populations and geographical boundaries [28,50,51]. This is of paramount importance, as some genetic variations are uncommon in certain racial groups but not in others. Most article clinical settings refer to cardiac surgery patients, critically ill patients and contrast-induced AKI, which is the third most common cause of AKI.

Chang et al. found that patients with contrast-induced AKI presented a higher decline in kidney function than those without contrast media [52]. What is more, Wu and coworkers confirmed in 2018 that baseline serum creatinine is an independent risk factor for contrast-induced AKI [31]. Men with baseline creatinine ≥114.9 µmol/L and women with creatinine ≥88.4 µmol/L are at a higher risk of developing this injury [53].

Understanding the contribution of environmental and genetic determinants to develop a disease represents one of the most significant challenges that researchers currently face. This is because the phenotype is shaped by genomes, the environment and their interactions. So, the response of genotypes to the environment is different for complex traits and diseases. Several studies have attempted to quantify the contribution of genetic and environmental risk factors to diseases [54,55,56]. It has been estimated that the environmental contribution to the appearance of a certain condition ranges from 13% to 90% [57,58]. Likewise, environmental factors can invoke heritable phenotype changes in DNA without alterations in its sequence. In this sense, epigenetics entails profound changes that affect gene activity and expression. Therefore, environmental determinants induce epigenetic marks that can trigger the development of certain illnesses.

## 2. Associated Genes

### 2.1. Polymorphisms

While an individual’s genotype depicts the combination of parental genotypes, two distinct individuals have >99.9% of their DNA sequences [59]. Variants observed in the left 0.1% of the human genome are known as gene polymorphisms and have become the topic of intense investigation. Indeed, such variations are markers of biological variety, and some genotypic polymorphisms have been identified to be related to particular human disease phenotypes [28]. It is unclear if any of these genetic polymorphisms are involved in the etiology of certain illnesses, as they may be placed close to other pathogenic genetic factors, known as linkage disequilibrium [28].

Variants can appear at one or more of the following locations: (1) the promoter region, (2) the exon(s) or the gene coding region, (3) the intron(s) or the gene intervening sequences and (4) the 3′-untranslated (3′-UTR) region (Figure 1).

Polymorphism of the promoter region may affect the transcriptional activity. Polymorphism of the exons or encoding regions may be mute or affect gene expression or function. As introns are transcribed but removed from the messenger RNA (mRNA) before it is translated into a protein, its polymorphism may cause defects in RNA and mRNA processing. Finally, polymorphism in the 3′-UTR region may alter the RNA expectancy or influence the mRNA ribosomal translation [28].

Three classes of gene variants have been reported: (1) single-nucleotide polymorphism (SNP), (2) variable number of tandem repeats (VNTRs) or minisatellite polymorphism and (3) microsatellite polymorphism, with SNP being the most common [28].

AKI susceptibility and severity are related to multiple genetic factors that are involved in several pathophysiological mechanisms as follows.

### 2.2. Systemic Inflammatory Response

Given the relevance of inflammatory processes in the development of AKI, polymorphisms in inflammation-related genes might influence the predisposition of an individual to AKI [20].

Among the important inflammation-related genes that could play a role in AKI are IL6, IL10, NFBK1, NFKBIA, IL18 and TNF.

Even rare polymorphisms with very low minor allele frequencies could provide vital information and potential usefulness as a marker in the investigation of genetic predisposition to AKI [60].

#### 2.2.1. Interleukin 6 (IL6)

*IL6* encodes for interleukin-6, which has been shown to induce a cellular-mediated immune reaction that causes kidney damage [61]. Nevertheless, there are conflicting results regarding IL6’s genetic polymorphism influence on AKI. While Dalboni et al. did not find any association between isolated or combined IL6 with other genetic polymorphisms and AKI development [62], Nechemia-Arbely et al. identified an association between IL6 and AKI development [63]. Furthermore, the plasma level of interleukin-6 has been found to serve as a biomarker for predicting AKI [64]. IL6-174 G/C polymorphism regulated postoperative IL6 levels and was related to the severity of postoperative AKI and length of hospital stay following coronary artery surgery [65]. Three promoter polymorphisms within the *IL6* genes, namely rs1800795, rs1800796 and rs1800797 polymorphisms, have been identified to influence the expression and secretion of the cytokine [66]. Stafford-Smith et al. [67]. Found that a combination of angiotensinogen (AGT) gene +842C allele (rs699) and IL-6–572C allele in Caucasians is related to kidney impairment.

#### 2.2.2. Interleukin 10 (IL 10)

The IL-10 gene is located on chromosome 1q31-32 [68], and the variation in IL-10 production is genetically set up and controlled at the transcriptional region [28]. The IL-10 promoter site is polymorphic with a single-base-pair replacement at position –1082 (G to A).

*IL10* encodes for interleukin-10, whose serum determination has also been associated with AKI [69,70]. Interleukin is implicated in AKI pathogenesis due to its anti-inflammatory role, as interleukin-10 facilitates the inhibition of immune cells and secretion of proinflammatory mediators, interrupting the healing process after kidney injury [70]. Promoter polymorphisms within the *IL10* gene, namely rs1800896 and rs3021097 polymorphisms, have been demonstrated to influence the level of the interleukin [20]. Low-producing genotype AA of IL 10 polymorphism rs1800896 has been related to AKI [71], along with the combined genotype of rs1800629 GG + rs1800896 AA [62]. Similarly, Hashad and colleagues have demonstrated that the low-producer genotype of IL-10 (–1082 G/A) variants was a predisposing factor for AKI in ICU patients with severe sepsis.

#### 2.2.3. Tumor Necrosis Factor-α (TNF-α)

The TNF-α gene is placed on the short arm of chromosome 6. Variants located in the promoter region of the TNF-α gene at positions −238 (G to A) and −308 (G to A) have been described. The −308 A allele, known as the TNF-α2 allele, increases promoter activity, boosts TNF·α production and has been related to superior serum creatinine and urinary kidney injury molecule-1 (KIM-1) levels and greater multiorgan failure calculations in patients with AKI [28,72].

TNF-α gene variants may alter variations in the proinflammatory cytokine reaction to stressful stimulation. This may have enormous implications in AKI presentation, as the intensity of proinflammatory reactions may determine the graveness of AKI and, therefore, the demand for renal replacement therapy and in-hospital mortality [28]. Figure 2 depicts an overview of the interstitial inflammation caused by TNF-α.

The TNF-α gene rs1800629 identified by Jaber and colleagues [28] is one of the most frequently studied polymorphisms in AKI [62,71,72,73,74,75,76]. It is related to superior levels of TNF-α in vitro, AKI predisposition and increased mortality in patients with renal replacement therapy (RRT) [52]. While the TNF-α gene rs361525 has not been associated with AKI [43,77], Hashad et al. demonstrated rs1800629 polymorphism is a predisposing factor for AKI in ICU patients with severe sepsis [71].

#### 2.2.4. Lymphotoxin α (LT-α) or Transforming Growth Factor β (TGF-β) and Interferon γ (IFN-γ)

IFN-γ is related to inflammatory response and renal damage [78]. Similarly, LT-α or TGF-β prompts neutrophils, T cells, monocytes and fibroblasts chemotaxis to the injury site [79]. Grabulosa and coworkers have shown that, although higher frequencies of polymorphisms of rs1800470, rs1800471 from the TGF-β and rs2430561 from IFN-γ were observed in critically ill patients, none was significantly associated as a risk factor for AKI [80].

#### 2.2.5. Human Leukocyte Antigen–Major Histocompatibility Complex, DR, B1 (HLA-DRB1)

HLA-DR is a major histocompatibility complex (MHC) class II cell receptor encoded on chromosome 6 region 6p21.31 [81]. It serves as a ligand for the T-cell receptor (TCR) and is involved in multiple autoimmune conditions. In addition, HLA-DR expression is an integral part of the glomerular capillary and peritubular endothelium [82]. In acute inflammation states, such as AKI, HLA-DR expression is exerted. Nevertheless, HLA-DRB alleles were found to be associated with less requirement of RRT [83].

#### 2.2.6. Nuclear Factor Kappa Beta 1 (NFKB1)

NFKB1 encodes for nuclear factor kappa beta 1 (NF-κB1), which is the most important member of the NF-κB family. Although it does not play a direct role in inflammation, it serves as the central regulator of a huge number of molecules involved in the inflammatory process. NF-κB1 functions as a central regulator for the activation and coordination of a vast assembling of genes involved in pro- and anti-inflammatory processes, including but not restricted to TNF, IL-1β and IL-6 [84]. NF-κB1 participates in the inflammation process via various signaling pathways; therefore, its related genes are intimately connected with the AKI pathogenesis [85,86,87,88].

The cellular level of NF-κB1 (also named p50 protein) is tightly controlled by IκBα, which is encoded by NFKBIA [89]. The rs2233406 and rs696 polymorphisms of the NF-BIA gene are, respectively, placed at the promoter and 3′UTR region of the gene.

*NFKB1* rs28362491, *NFKBIA* rs2233406 and *NFKBIA* rs696 polymorphisms were related to reduced predisposition of AKI among Chinese children [20]. The NFKB1 rs28362491 polymorphism is an insertion/deletion variation of the gene. This insertion decreases the binding affinity of the promoter sequence and leads to a reduced NFKB1 promoter activity, which results in a low inflammation activity. NFKBIA rs2233406 polymorphism, however, occurs at the promoter region, and the variant T allele may disrupt the GATA-2 transcription factor binding, leading to a decreased transcriptional activity of the gene [89]. Conversely, the variant G allele of the NFKBIA rs696 polymorphism can enhance the gene transcription by reducing the binding affinity of miR-449a microRNA on the gene. This exemplifies the complexity of the interactions by which genetic polymorphisms could affect disease susceptibility.

#### 2.2.7. Macrophage Migration Inhibitory Factor (MIF)

Macrophage migration inhibitory factor (MIF) is a cytokine implicated in various inflammatory processes that is rapidly released from preformed intracellular pools in response to multiple cellular and systemic noxious stimuli, including ischemia/reperfusion, endotoxemia and surgery. For instance, cardiac surgery generates an increase in MIF serum levels [90].

Averdunk et al. showed that macrophage migration inhibitory factor (MIF) promoter polymorphisms (rs3063368, rs755622) are associated with AKI and death after cardiac surgery [91]. MIF may mediate AKI via CD74/TLR4-NF-KB pathway [92].

#### 2.2.8. Interleukin-18 (IL-18)

Interleukin-18, encoded by *IL18*, is also implicated in AKI pathogenesis. Studies have shown that interleukin-18 is linked to AKI, inducing kidney acute tubular necrosis [93,94]. Thus, a disrupted level of interleukin-18 could serve as a risk factor for AKI. Promoter polymorphisms in the *IL18* gene may influence the level of the cytokine. Two such *IL18* polymorphisms are the rs1946518 and rs187238 polymorphisms.

### 2.3. Vascular Hemodynamic Response

#### 2.3.1. Vascular Endothelial Growth Factor (VEGF)

Vascular endothelial growth factor (VEGF) is a protein that promotes angiogenesis, vessel permeability, cellular survival and differentiation [95,96]. The rs3025039 genotype has been shown to boost AKI predisposition in critically ill patients with severe sepsis [73].

#### 2.3.2. Angiotensinogen (AGT) and Angiotensin-Converting Enzyme (ACE)

The AGT gene is located on chromosome 1 band q42. It encodes the angiotensinogen precursor, which is a fundamental component of the renin–angiotensin–aldosterone system (RAAS), being a potent vasoconstrictor. It is a major regulator of blood pressure, fluid and electrolyte homeostasis, playing a key role in renal disease pathology (Figure 3). While AGT 842C was identified by Stafford-Smith [67] as a major risk factor to develop postoperative renal injury, Isbir et al. [97]. Found that AGT receptor 1 (AGTR1) does not have a single relationship with AKI.

On the other hand, the angiotensin-converting enzyme (ACE) gene is placed on chromosome 17 band q23. Several studies have analyzed ACE insertion/deletion (I/D) polymorphism (rs4646994) [98,99,100,101]. Isbir and coworkers [97] found a relationship between the ACE D allele and an augmented susceptibility of postoperative AKI after coronary artery bypass graft surgery. Conversely, du Cheyron and colleagues [100] identified that I/I genotype is related to a higher vulnerability to AKI and RRT.

#### 2.3.3. Endothelial No Synthase (eNOS)

eNOS is the principal responsible for the vascular generation of nitric oxide, hence its variants might play a crucial role in the pathogenesis of endothelial dysfunction. Popov et al. demonstrated that T-786C eNOS polymorphism may predispose renal dysfunction and increase the incidence of dialysis following cardiac surgery [102]. Likewise, Stafford and coworkers showed that 894T eNOS polymorphism was associated with cardiac surgery-associated AKI (CSA-AKI) in Caucasians, as this variant may increment vascular tone and contribute to medullary ischemia [67].

### 2.4. Cellular Metabolic Homeostasis

#### 2.4.1. Cytochrome b_245_

Cytochrome b245 is involved in phagocytosis. While the A allele of the rs8854 polymorphism in the Cytochrome b245 α subunit (CYBA) gene was associated with less renal replacement therapy and hospital death compared to the GG genotype, the haplotype A-A-G-G of polymorphisms rs4782390, rs4673, rs3794624 and rs8854 was related to a higher vulnerability to this outcome [103].

#### 2.4.2. Kallikrein-1 (KLK1)

Kallikreins belong to a family of serine proteases with diverse physiological functions. Kallikrein-1 (KLK1), one of 15 kallikrein family members, is the principal kallikrein expressed in the kidney and is implicated in both renal function and blood pressure control via vasodilatory and natriuretic results [104].

KLK1 is encoded by the KLK1 gene, which is located on chromosome 19q13.3 [104]. Polymorphisms at the promoter region have been identified and associated with CKD [105,106]. Susantitaphong and colleagues found that the I and G alleles of the KLK1 promoter polymorphism were related to an increased risk for AKI severity, including a doubled increase in serum creatinine, oliguria and RRT [104].

#### 2.4.3. Nicotinamide Adenosine Dinucleotide Phosphate (NADPH)

Perianayagam et al. [107] found that the rs4673 variant in the gene that encodes the NADPH oxidase p22phox subunit at position +242 is related to RRT and death among patients with AKI.

#### 2.4.4. PH Domain and Leucine-Rich Repeat Protein Phosphatase 2 (PHLPP2)

PH domain and leucine-rich repeat protein phosphatase 2 (PHLPP2) is a phosphatase essential for the control of PKC isoforms and Akt kinases [108,109]. Akt kinases, known as prosurvival kinases, regulate the equilibrium between cell survival and apoptosis, aside from proliferation and cellular quiescence. The rs78064607 polymorphism located in the PHLPP2 gene was the only SNP identified by Westphal et al. [110] in their genome-wide study with increased risk for AKI.

### 2.5. Adrenergic Response

AKI produces the synthesis and release of catecholaminergic hormones as a response to any acute physiologic stress. As such, this pathway may play a relevant role in the etiology, pathogenesis, evolution and outcome of the disease.

#### 2.5.1. Catechol-O-Methyltransferase Gene (COMT)

The catechol-O-methyltransferase gene (COMT) encodes the COMT enzyme, which degrades catecholamines and contributes to vasodilatory shock and AKI [111]. Albert et al. and Haase-Flelitz et al. found that the low activity (L) rs4680 polymorphism or LL genotype is related to CSA-AKI, more furosemide administration and RRT [111,112]; however, this association was discarded by Kornet et al. [113] and Albert et al. [112].

#### 2.5.2. Phenylethanolamine N-Methyltransferase (PNMT)

PNMT is the final product of the catecholaminergic pathway and converts noradrenaline to adrenaline. While the gene rs5638 + 1543 G allele is related to an augmented predisposition for AKI, and the genotype +1543 G/A is related to oliguria, the PNMT rs876493-161 A allele is associated with diminished mortality and less circulatory collapse [114].

### 2.6. Cell Proliferation and Differentiation

#### 2.6.1. Glutamate Receptor Metabotropic 7 (GRM7) and LMCD1 Antisense RNA 1 (LMCD1-AS1)

GRM7 encodes a protein G-coupled receptor and is one of the group III metabotropic glutamate receptors, which are linked to the inhibition of the AMPc cascade. LMCD1-AS1 is a recognized oncogene and exerts a proliferation function.

The intergenic region *GRM7*|*LMCD-AS1* located at chromosome 3p21.6 was detected to be highly related to CSA-AKI by Stafford-Smith et al., although no direct functional role was currently attributed to this intergenic region [115]. Future studies are needed to confirm this finding.

#### 2.6.2. Salt-Inducible Kinase 3 (SIK3)

The SIK family members, including SIK3, are serine/threonine kinases that belong to the AMP-activated protein kinase (AMPK) family [116,117]. As their activity is upregulated in various cancers, they might play a key role in tumor appearance and progression. Polymorphism rs625145 in the SIK3 gene was correlated to elevated risk for AKI in patients with septic shock [118].

#### 2.6.3. Suppressor of Fused Homolog (SUFU)

Hedgehog signaling is one of the crucial regulators of cell differentiation and has been deeply studied in the context of cancer. It also plays a key role in immune activation and inflammation, and its expression is upregulated while damaged organs are being healed. A key negative regulator of this signaling process is the suppressor of fused homolog (SUFU).

The polymorphisms rs10786691, rs12414407, rs10748825 and rs7078511 in the SUFU gene have been associated with renal performance in ICU patients with Enterobacteriaceae sepsis [119].

#### 2.6.4. Transducer and Activator of Transcription 3

Transducer and activator of transcription 3 (STAT3) is a controller of T-helper 17 (Th17 cells) differentiation and function. Furthermore, STAT3 regulates and responds to various cytokines, including IL-1β, IL-10, IL-6, IL-8, IL-11, IL-17, IL-21 and IL-23 [120].

The STAT3 rs1053004 polymorphism was significantly related to a lower vulnerability of CSA-AKI in an older Iranian population [121].

#### 2.6.5. Erythropoietin (EPO)

While no association with AKI was found, Popov identified that the EPO gene rs1617640 T/G-polymorphism TT genotype was related to a higher creatine phosphokinase-MB (CPK-MB) and RRT requirement [122].

#### 2.6.6. Surfactant Protein-D (SP-D)

The SP-D gene is located in chromosome 10q22.2-q23.1 and is expressed in kidney tubules [123,124]. Liu et al. [125] demonstrated that patients with SP-D-11Thr/Thr genotype were more prone to AKI compared to those with other SP-D genotypes in a Chinese population. They showed that the more severely damaged the renal epithelial cells, the more SP-D protein will escape into the bloodstream, explaining the relationship between SP-D levels and AKI severity. In fact, greater serum SP-D levels were related to adverse clinical outcomes, such as higher AKI stage, longer RRT and increased mortality [125].

### 2.7. Lipid Metabolism

#### 2.7.1. Adiponectin

Adiponectin is a multifaceted cytokine that has a major role in the adjustment of energy metabolism and inflammatory response [126,127]. While initial reports informed that they are exclusively encountered in adipocytes [128], recent research has demonstrated that they are also produced by lymphocytes [129], macrophages and endothelial and epithelial cells [130]. Bloodstream adiponectin levels are higher in patients with chronic kidney disease (CKD), and elevated levels of adiponectin predict increased cardiovascular and all-cause mortality and CKD progression [131].

Jin et al. [132] showed that adiponectin plays a pivotal role in the pathogenesis of acute renal ischemia/reperfusion injury and may be a potential therapeutic target.

#### 2.7.2. Apolipoprotein E (APOE)

Apolipoprotein E is a protein implicated in lipid homeostasis, tissue restoration and immune response. The gene is located on chromosome 19q13.2 [44]. The polymorphism rs7412 of apolipoprotein E (APOE) and the non-e4 allele of polymorphism rs429358 are related to a greater peak serum creatinine [133,134] and an increased risk of postoperative AKI [97,134].

#### 2.7.3. Apolipoprotein L1 (APOL1)

Variations in apolipoprotein L1, encoded by the APOL1 gene, are predisposing factors for focal segmental glomerulosclerosis, chronic renal and end-stage renal disease in African Americans [135,136], which may be related to renal blood flow impairment.

IRF2 controls the expression of the kidney disease risk gene APOL1. Zhao and colleagues found that rs62341639 and rs62341657 polymorphism on chromosome 4 near the APOL1 regulator IRF2 and rs9617814 and rs10854554 polymorphism on chromosome 22 close to the acute kidney injury-related gene TBX1 were associated with AKI development but without genome-wide significance [137]. TBX1 is a T-box transcription factor implicated in embryonic renal development, which was associated with the stimulation of TGF-β and with renal damage in a gentamicin-induced AKI study [138].

### 2.8. Noxious Stimuli

#### Hypoxia-Inducible Factor-1-α

Hypoxia-inducible factor-1-α controls the cell response to hypoxic conditions. In 2009, Kolyada et al. [139] found that the T allele of the rs11549465 variant located on the transcription factor of the hypoxia-inducible factor-1alpha (HIF-1α) gene is associated with the increased need for RRT.

### 2.9. Oxidative Stress

#### 2.9.1. Myeloperoxidase (MPO)

Myeloperoxidase is a lysosomal enzyme involved in oxidative stress and helps granulocytes destroy pathogens. In 2012, Perianayagam found that rs2243828, rs2071409, (rs2759) and rs7208693 polymorphisms of the MPO gene were associated with lower urine output, more dialysis requirement and higher in-hospital mortality [140].

#### 2.9.2. Catalase

Oxidative stress plays a major role in the pathogenesis of multiple conditions, and AKI is not an exception. The balance between the formation of reactive oxygen species and antioxidants is determinant. The T allele of the rs769217 variant of the antioxidant defense enzyme catalase (CAT) gene was associated with hospital morbidity and death among AKI patients in a Turkish population [141].

### 2.10. Cell Survival and Apoptosis

#### 2.10.1. B-Cell CLL/Lymphoma 2 (BCL2)

BCL2 has a major part in apoptosis signaling as an antiapoptosis protein [142]. Carriers having the minor alleles of rs8094315 and rs12457893 polymorphism on BCL2 had a declined predisposition for developing AKI [118].

#### 2.10.2. Serpin Peptidase Inhibitor (SERPIN)

SERPIN clade A (alpha-1 antiproteinase, antitrypsin) member 4 (SERPINA4) gene encodes kallistatin, which has vasodilatory, antioxidant, anti-inflammatory, and antiapoptotic properties [143]. Kallistatin suppresses TNF-α induced-apoptosis in in vitro endothelial cells [144]. Thus, polymorphism rs2093266 in the SERPINA4 gene was found to protect patients with septic shock against AKI [118].

### 2.11. Cytoskeleton

#### Bardet–Biedl Syndrome 9 BBS9

Bardet–Biedl syndrome 9 (BBS9) gene plays a fundamental role in the control of cilia length through the adjustment of actin cytoskeleton polymerization [145]. Stafford-Smith and colleagues [115] in their GWAS identified a relationship of polymorphism rs10262995 in BBS9 with CSA-AKI. Nevertheless, Vilander et al. did not find this association [47].

The association between gene studies and AKI susceptibility is depicted in Figure 4.

## 3. Conclusions

The overview of AKI pathogenesis suggests that various genes act collectively, generating either a favorable or harmful environment of pro- and anti-inflammatory cytokines, which determines the intensity of tissue damage. From the review, the most relevant genes are those related to the systemic inflammatory response, especially TNF-α, which illustrates the huge impact that this process has on the etiology and pathophysiology intricacy of this disease. Genetic variants may influence the kidney response to an injury, deciding whether a patient moves towards a more serious condition or recovery. Thus, genetic variety provides a useful rational approach as clinical risk factors, unfolding partially of the global risk.

Examining the association between polymorphisms and AKI risk could potentially provide important insights into the disease. As it can help identify susceptible patients based on genotype patterns with the aim to prevent or ameliorate kidney damage. A further understanding of these genetic variants can serve to develop a substantial improvement to tackle this disease through the development of novel risk stratification scores and new genetic specific targets. For instance, even though novel biomarkers, such as neutrophil gelatinase-associated lipocalin (NGAL), KIM-1, tissue inhibitors of metalloproteinases 1 (TIMP-1) and insulin-like growth factor binding protein 2 (IGFBP2), have been identified for the early detection and diagnosis of AKI. The relationship between the genetic variants of these markers with AKI predisposition has not been determined yet. This may be a wise approach to further tackle this disease. Thus, the knowledge of genetic determinants could benefit patient management as it can lend a hand to the creation of a genetic risk stratification tool with the development of genetic susceptibility biomarkers.

## Figures and Tables

**Figure 1 jcm-10-03039-f001:**
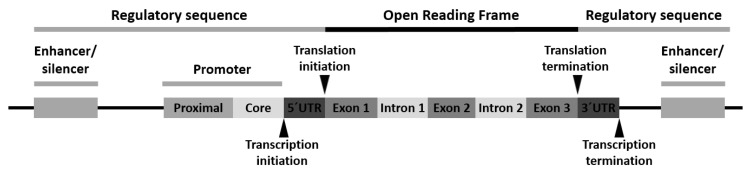
Schematic arrangement of a typical human gene with locations for gene polymorphism.

**Figure 2 jcm-10-03039-f002:**
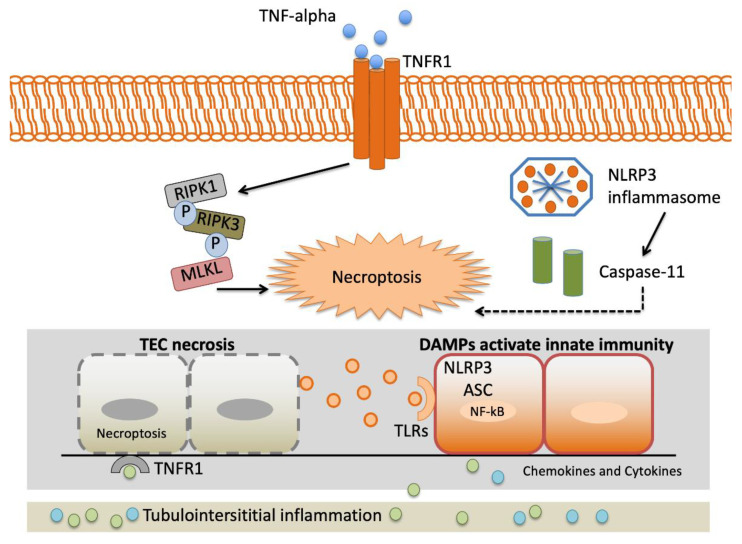
Landscape of interstitial inflammation caused by TNF-α. Complexity of the TNF-α signaling system. TNFR1 signaling results in both activation of proinflammatory pathways and necroptosis triggered by the RIPK3-MLKL system. The NLRP3 inflammasome is a crucial element of the innate immune system that favors caspase activation and the production of proinflammatory cytokines due to damage-associated molecular patterns (DAMPs). Abbreviations: Apoptosis-associated speck-like protein containing a caspase recruitment domain (ASC). Damage-associated molecular patterns (DAMPs). Leucine-rich repeat-containing protein 3 (NLRP3). Mixed-lineage kinase domain-like (MLKL). Nuclear factor-kappa light-chain enhancer of activated B cells (NF-κB). Receptor interacting protein kinase-1 (RIPK1). Receptor interacting protein kinase-3 (RIPK3). Tubular epithelial cell (TEC). TNF receptor 1 (TNFR1). Toll-like receptor (TLR).

**Figure 3 jcm-10-03039-f003:**
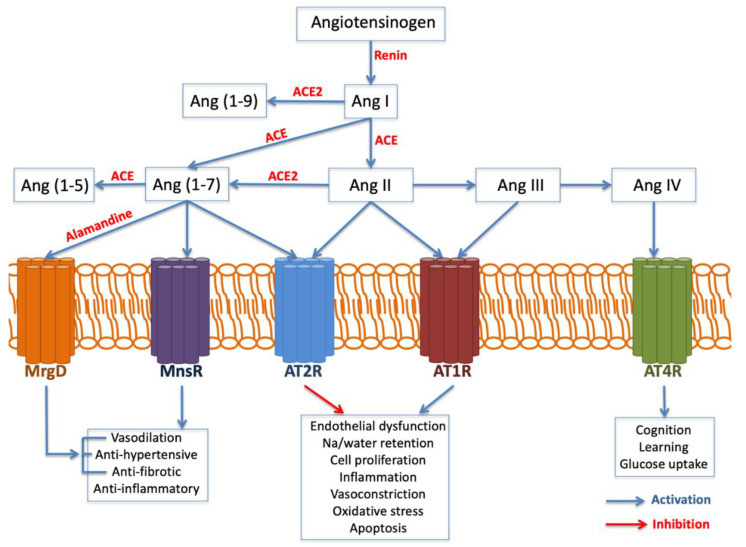
Angiotensin generated by angiotensin-converting enzyme (ACE) exerts inflammation, oxidative stress, apoptosis and fibrosis. Angiotensin II, through AT1R, is a prominent activator of endothelial dysfunction, cell proliferation, inflammation, vasoconstriction and oxidative stress mediated by the NADPH oxidase complex, with the production of oxygen reactants, which contribute to the oxidative stress and inflammatory processes involved in kidney injury. Abbreviations: Angiotensin (Ag). Angiotensin receptor (ATR). Mas-related G-protein-coupled receptor type D (MrgD).

**Figure 4 jcm-10-03039-f004:**
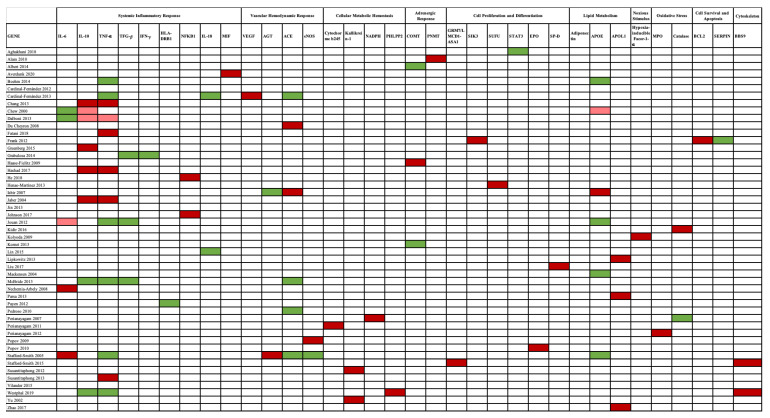
Genes and AKI susceptibility relationship. A red color pinpoints the genes associated with AKI, in which a dark red color highlights a strong association, while a light red color indicates a weak association. No association is marked by a green color. Genes are presented according to the mechanism involved in AKI pathogenesis. Abbreviations: ACE, angiotensin-converting enzyme gene; AGT, angiotensinogen gene; APOE, apolipoprotein E gene; BCL-2, B-cell CLL/lymphoma 2 gene; BBS9, Bardet–Biedl syndrome 9 gene; CAT, catalase gene; COMT, catechol-O-methyltransferase gene; eNOS, endothelial nitric oxide synthase gene; EPO, erythropoietin gene; HIF1-α, hypoxia-inducible factor 1-α gene; HLA-DRB1, human leukocyte antigen–major histocompatibility complex, class II, DR beta 1 gene; IFN-γ, interferon-γ gene; IL-6, interleukin-6 gene; IL-18, interleukin-18 gene; IL-10, interleukin-10 gene; MIF, macrophage migration inhibitory factor gene; MPO, myeloperoxidase gene; NADPH, nicotinamide adenosine dinucleotide phosphate gene; NFκB, nuclear factor-κB gene; PHLPP2, PH domain and leucine-rich repeat protein phosphatase 2 gene; PNMT, phenylethanolamine N-methyltransferase gene; SERPINA, serpin peptidase inhibitor, clade A (α-1 antiproteinase, antitrypsin) gene; SIK3, salt-inducible kinase 3 gene; SP-D, surfactant protein-D gene; STAT3, transducer and activator of transcription 3 gene; SUFU, suppressor of fused homolog gene; TGF-β, transient growth factor-β gene; TNF-α, tumor necrosis factor α gene; VEGF, vascular endothelial growth factor gene.

**Table 1 jcm-10-03039-t001:** AKI susceptibility studies with their investigated gene polymorphisms along with their related evidence presented chronologically.

Study	Design	Clinical Settings	Participants/Population Characteristics	Patients with AKI	Studied Polymorphisms	Outcomes	AKI Definition
**Chew, et al.,** **2000**	Prospective, Observational Cohort Study	CSA-AKI	564 patients undergoing CABG Irish population	---	Apolipoprotein E (APOE)	APO4 ε4 allele is associated with reduced postoperative increase in serum Cr after cardiac surgery, compared with the ε3 or ε2 allele	By comparisons of preoperative (CrPre), peak in-hospital postoperative (CrMax) and perioperative change (∆Cr) in serum Cr values
**Yu, et al.,** **2002**	Case/Control Study	End-stage renal disease (ESRD) in African Americans	85 control subjects, 92 type 2 diabetes ESRD patients, and 76 non-diabetic ESRD families. US population	199 Patients	KLK1 promoter	KLK1 is associated with hypertension ESRD	
**Jaber, et al.,** **2004**	Prospective Study	In-hospital patients requiring dialysis	Hospitalized patients who required intermittent hemodialysis England population	61 Patients	TNF-α and IL-10	TNF-α and IL-10 gene polymorphisms are related to increased mortality among patients with AKI requiring dialysis	Renal failure requiring dialysis
**Mackensen, et al.,** **2004**	Prospective, Observational Cohort Study	Elective CABG patients	130 coronary patients US population	---	Apolipoprotein E (APOE)	Non-APOE4 patients are more vulnerable to AKI after cardiac operation APO4 ε4 allele is associated with a nephroprotective effect	Perioperative difference in serum Cr (∆Cr = Crmax Post – Cr Pre)
**Stafford-Smith, et al.,** **2005**	Prospective, longitudinal study	CSA-AKI	3149 patients undergoing CABG US population	More than half the patients	Interleukin 6 -572C, AGT 842C, APO E 448C [ε4], AGTR1 A1166C, and [eNOS] 894T ACE I/D	AGT 842C and IL-6 ---572C, a variant pattern that occurs in 6% of Caucasians) was associated with major postoperative renal injury, with an average peak serum Cr increase of 121%	Difference between preoperative and peak postoperative Cr values
**Isbir, et al.,** **2007**	Prospective, Observational Cohort Study	CSA-AKI	248 elective CABG patients Turkish population	54 Patients	ACE II, ID, and DD, APO E, and AGTR1 A1166C genotype	ACE I/D and APO E gene polymorphisms may play a role in the development of AKI after cardiac surgery. AGTR1 does not have a unique association with postoperative AKI	RIFLE
**Perianayagam, et al.,** **2007**	Prospective Cohort Study	Hospitalized patients with AKI	US population Hospitalized patients with established AKI of mixed cause and severity	200 patients	Coding region (C to T substitution at position +242) NADPH oxidase p22phox subunit gene and promoter region (C to T substitution at position -262) of the catalase gene	NADPH oxidase p22phox subunit at position +242 is associated with dialysis requirement or hospital death among patients with AKI	An increase in Cr by 0.5, 1.0, or 1.5 mg/dL from a baseline level of ≤1.9, 2.0 to 4.9, and ≥5.0 mg/dL
**Nechemia- Arbely, et al.,** **2008**	Animal Basic Research	Experimental mice model	IL-6–deficient mice were compared with wild-type mice	---	IL-6/sIL-6R	IL-6 promotes a renal inflammatory response	Nephrotoxic-induced AKI
**Du Cheyron, et al.,** **2008**	Prospective Cohort Study	AKI in ICU admitted patients	180 ICU patients France population	73 patients	ACE I/D polymorphism	ACE II genotype was independently associated with increased risk of AKI	RIFLE
**Haase-Fielitz, et al.,** **2009**	Prospective, observational, cohort study	CSA-AKI	260 patients undergoing cardiac surgery Australian population	53 patients	Catechol-O-methyltransferase (COMT)	COMT LL homozygosity is an independent risk factor for AKI	RIFLE
**Popov, et al.,** **2009**	Prospective Study	CSA-AKI	497 patients undergoing cardiac surgery German population	287 patients	T-786C endothelial NO synthase (eNOS)	T-786C eNOS polymorphism is associated with AKI and increase the occurrence of RRT following cardiac surgery	RIFLE
**Kolyada, et al.,** **2009**	Prospective Cohort Study	Hospitalized patients with AKI	Adult patients with AKI US population	241 patients	Hypoxia-inducible factor-1a (HIF-1α)	HIF-1 α gene polymorphism predicts adverse outcomes in hospitalized patients with AKI	AKIN
**Alam, et al.,** **2010**	Case/Control Study	Hospitalized patients with AKI	961 Caucasian subjects US population	194 Patients	Phenylethanolamine N-methyltransferase (PNMT) PNMT promoter G–161A (rs876493) and coding A + 1543G (rs5638)	In Caucasians, PNMT SNPs are associated with the development of AKI, disease severity, and in-hospital mortality	AKIN
**Popov, et al.,** **2010**	Prospective Cohort Study	CSA-AKI	481 Patients undergoing Cardiac Surgery German population	274 Patients	SNP rs1617640 in the promoter of the EPO gene	EPO rs1617460 TT allele is associated with more acute RRT	RIFLE
**Perianayagam, et al.,** **2011**	Prospective, Observational Cohort Study	Hospitalized adults with AKI	256 hospitalized patients US population	256 Patients	CYBA gene polymorphisms (rs8854, rs3794624, rs4673, rs4782390, and rs1049255)	*CYBA* rs4782390, rs4673, rs3794624, and rs8854 polymorphisms were associated with dialysis requirement	A rise in serum Cr by 0.5, 1.0, or 1.5 mg/dL from a baseline level ≤1.9, 2.0–4.9, or ≥5.0 mg/dL
**Jouan, et al.,** **2012**	Prospective Cohort Study	CSA-AKI	126 Patients undergoing CABG France population	8 Patients	LTA (Cys13Arg,þ252A > G), TNF-α (-308G > A), IL6 (-597G > A, -572G > C, -174G > C), IL10 (-592C > A, c.*117C > T), and APOE (Cys112Arg, Arg158Cys).	IL6-572GCþCC/IL10-592CC were associated with AKI	As Cr levels > 200 mmol/L or, particularly for patients having a baseline plasma Cr level > 150 mmol/L, the requirement for dialysis at any time after surgery.
**Cardinal-Fernandez, et al.,** **2012**	Systematic Review	Genetic Predisposition to AKI	4.835 patients included	12 References	ACE, eNOS, FNMT y COMT, TNF-α, IL10, IL6, HIP-1A, EPO, NAPH oxidase, and APOE	AKI susceptibility and severity is related to genetic factors that are involved in different physiopathological mechanisms	AKI term search
**Susantitaphong, et al.,** **2012**	Case/Control Study	Hospitalized cases with AKI of multiple etiology from two acute care facilities	481 subjects (214 hospitalized patients with AKI of mixed causes and 267 healthy subjects)	214 Patients	Multiallelic KLK1 promoter gene	KLK1 promoter polymorphisms are associated with development of AKI and adverse outcomes	AKIN
**Perianayagam, et al.,** **2012**	Prospective Cohort Study	Hospitalized patients with AKI	262 adults hospitalized with acute kidney injury	262 Patients	MPO polymorphisms rs2243828, rs7208693, rs2071409, and rs2759	MPO polymorphisms rs2243828, rs7208693, rs2071409, and rs2759 were associated with lower urine output, more dialysis requirement and higher in-hospital mortality.	AKIN
**Payen, et al.,** **2012**	Prospective multicenter observational study	Critical ill patients with severe sepsis and septic shock	221 Patients France population	129 Patients	HLA-DRB1	HLA-DRB alleles were found to be associated with less requirement of RRT	AKIN
**Frank, et al.,** **2012**	Retrospective Study	AKI in Patients with septic shock	1,264 patients with septic shock	637 Patients	BCL2 Genetic Variants SERPINA4 SNP rs2093266	BCL2 SNPs rs8094315 and rs12457893 were associated with a decreased risk of developing AKI SERPINA4 SNP rs2093266 was linked to a decreased risk to develop AKI	AKIN
**Jin, et al.,** **2013**	Animal Basic Research	Mouse Model of Ischemic-Reperfusion Injury	Wild-type mice, compared to adiponectin knockout mice	---	Adiponection	Adiponectin wild-type mice had less kidney dysfunction and tubular damage. There was more inhibition of NF-κB activation and reduced expression of the proinflammatory molecules IL-6, TNF-α, MCP-1, and MIP-2	Ischemic-Reperfusion Injury
**Cardinal-Fernández, et al.,** **2013**	Prospective, observational, cohort study	Patients admitted to the ICU with severe sepsis	139 Patients with severe sepsis	65 patients	Angiotensin-converting enzyme insertion/deletion; tumor necrosis factor α−376, −308, and −238; interleukin-8 − 251; vascular endothelial growth factor (VEGF) +405 and +936; and pre–B-cell colony-enhancing factor −1001	VEGF + 936 CC genotype increased the risk to develop AKI in patients with severe sepsis.	RIFLE
**McBride, et al.,** **2013**	Prospective, observational, cohort study	CSA-AKI	408 elective cardiac surgery patients Irish population	69 Patients	TNF/G-308A, TGF-β 1-509 C/T, IL10/G-1082A and ACE I/D.	TNF/G-308A, TGF-β 1-509 C/T, IL10/G-1082A and ACE I/D. genotype were not associated with AKI	Drop from baseline eGFR of greater than 25% (as calculated by the method of MDRD.
**Susantitaphong, et al.,** **2013**	Cohort Study	Hospitalized patients with AKI were recruited from two acute care hospitals	262 hospitalized German population	262 Patients	Promoter region of TNF-α	The TNF-α rs1800629 gene polymorphism is associated with markers of kidney disease severity and distant organ dysfunction among patients with AKI	A rise in serum Cr by 0.5, 1.0, or 1.5 mg/dL from a baseline level of ≤1.9, 2.0–4.9, or ≥5.0 mg/dL
**Chang, et al.,** **2013**	Prospective Case/Control Study	Patients who underwent coronary artery intervention	53 contrast induced-AKI patients compared to 455 control subjects.	53 contrast induced-AKI patients	Four IL-10 tag SNPs (rs1554286, rs3021094, rs3790622, rs1800896) and three TNF-α tag SNPs (rs1799964, rs1800630, rs1800629)	Gene polymorphisms of IL-10 and TNF-α are associated with Contrast induced-AKI	A rise in Cr of ≥0.5 mg/dL (44 mmol/L) or a 25% increase from baseline value, assessed within 48 h after a radiological procedure
**Kornet, et al.,** **2013**	Prospective, Observational Cohort Study	CSA-AKI	1741 patients undergoing elective cardiac surgery German population	398 Patients	COMT-Val158Met-(G/A) polymorphism (rs4680)	COMT-Val158Met-(G/A) polymorphism (rs4680) was not associated with CSA-AKI	RIFLE
**Dalboni, et al.,** **2013**	Prospective nested case–control	ICU Setting	303 ICU patients and 244 healthy individuals	139 Patients	-308 G < A (TNF)- α, -174 G > C IL-6 and -1082 G > A IL-10	Both low TNF-α and low IL-10 producer phenotypes were an independent risk factor of AKI and/or death and RRT and/or death in ICU patients.	AKIN, and RIFLE
**Lipkowitz, et al.,** **2013**	Case/Control Study	Hypertension-attributed nephropathy who developed severe CKD	675 Cases compared to 618 Controls African Americans	675 Patients	APOL1 and MYH9 genes	APOL1 risk variants were consistently associated with renal disease progression	(1) Developing ESRD or Cr > 2 mg/mL; (2) developing ESKD or Cr > 3 mg/dL
**Parsa, et al.,** **2013**	AASK and CRIC Study Cohort Study	Black patients in the United States with chronic kidney disease US Population	AASK: 693 black patients with chronic kidney disease attributed to hypertension CRIC: 2955 white patients and black patients with chronic kidney disease	492 Patients in the AASK Study	APOL1	APOL1 were associated with the higher rates of end-stage renal disease and progression of CKD that were observed in black patients as compared with white patients	A doubling of the Cr (equivalent to a reduction of 50% in the GFR) from baseline or incident ESRD
**Henao-Martínez, et al.,** **2013**	Prospective, Observational Cohort Study	AKI in sepsis	250 hospitalized patients	159 Patients	SUFU	rs10786691 (*p* = 0.03), rs12414407 (*p* = 0.026), rs10748825 (*p* = 0.01), and rs7078511 correlated to AKI	Based on Serum Cr
**Boehm, et al.,** **2014**	Prospective, Observational Cohort Study	CSA-AKI	1415 elective cardiac surgery patients German population	318 Patients	Apolipoprotein E (ApoE ε2, ε3, ε4) (rs429358 and rs7412) and TNF-α-308 G > A (rs1800629).	ApoE (ε2, ε3, ε4) polymorphism and the TNF-α-308 G > A polymorphism are not associated with CSA-AKI	RIFLE
**Grabulosa, et al.,** **2014**	Prospective nested case-control study	ICU patients	139 ICU AKI patients, 164 ICU patients without AKI, compared to 244 healthy individuals.	139 Patients	rs1800470 (codon 10 T/C), rs1800471 (codon 25 C/G) from the TGF-β, and rs2430561 (+874 T/A) from IFN-γ	Genetic polymorphism of the TGF-β and IFN-γ was not associated as a risk factor for AKI	AKIN, RIFLE
**Albert, et al.,** **2014**	Prospective, Observational Cohort Study	CSA-AKI	195 patients German Cohort	22 Patients	COMT-Val158Met	COMT genotype may associate with different patterns of renal functional changes and tubular stress biomarker response after cardiac surgery.	RIFLE
**Lin, et al.,** **2015**	Systematic Review	AKI	11 References from 3 countries 2796 patients	538 patients	IL-18	IL-18 could be used as a biomarker in the prediction of AKI	RIFLE, AKIN, pRIFLE
**Stafford-Smith, 2015**	GWAS	AKI following CAGB	873 non-emergent CABG patients (discovery) 380 cardiac surgery patients (replication) US population	294 in Discovery Cohort 119 in Replication Cohort	The rs13317787 in GRM7|LMCD1-AS1 intergenic region (3p21.6) and rs10262995 in BBS9 (7p14.3)	GRM7|LMCD1-AS1 and BBS9 were associated with post-CABG AKI	KDIGO, AKIN, RIFLE
**Greenberg, et al.,** **2015**	Prospective Study	Pediatric Cardiac Surgery	Cohort, including 106 children ranging in age from 1 month to 18 years undergoing CPB	24 Patients	IL-6 and IL-10	Preoperative plasma IL-6 levels are associated with AKI	At least a doubling of the baseline Cr concentration or dialysis
**Vilander, et al.,** **2015**	Systematic Review	Genetic predisposition to AKI	4027 References	28 References	ACE; AGTR1; AGT; APOE; BCL-2; COMT; CYBA; eNOS; EPO; FCGR2A; FCGR3A; FCGR3B; GLI1; HHIP; HIF-1- α; HLA-DRB1; IL-6; IL-8; IL-10; LTA; MPO; NADPH; PBEF; PNMT; PTCH1; PTCH2; SERPINA4; SERPINA5; SIK3; SMO; SUFU; TGF-β; TNF-α; VEGF.	Articles quite heterogeneous and of moderate quality	KDIGO, AKIN, RIFLE
**Kidir, et al.,** **2016**	Cross-sectional Study	Case (AKI)/Control Hospitalized patients	Turkish population 90 AKI patients compared to 101 healthy volunteers	90 AKI patients	MnSOD rs4880, GPX1 rs10500450 and CAT rs769217	T allele of CAT rs769217 was associated with increased morbidity and mortality	KDIGO
**Zhao, et al.,** **2017**	GWAS	Cases and controls for the discovery population were derived from two independent populations of critically ill patients. The second population enrolled patients who underwent cardiac surgery	Discovery population: 760 acute kidney injury cases and 669 controls. Replication population: 206 cases and 1406 Controls	760 patients (Discovery) and 206 (Replication)	APOL1-regulator IRF2 and AKI–related TBX1 genes	rs62341639 and rs62341657 on chromosome 4 near APOL1-regulator IRF2, and rs9617814 and rs10854554 on chromosome 22 near acute kidney injury–related gene TBX1 are associated with AKI	At least 0.3-mg/dL or 50% increase in Cr from baseline
**Johnson, et al.,** **2017**	Animal Basic Research	Experimental mice model	74 male Wistar rats	74 mice	Nuclear factor-κB (NFκB)	Inhibition IκB kinase improves kidney recovery and decreases fibrosis	AKI caused by unilateral nephrectomy plus contralateral ischemia and reperfusion injury
**Hashad, et al.,** **2017**	Prospective Cohort Study	Critical ill patients with severe sepsis	150 patients with severe sepsis	66 patients	-398 G/C of TNF-α and -1082G/A of IL-10	Genotypes of both TNF-α and IL-10 were associated with AKI	---
**Liu, et al.,** **2017**	Case/Control Study	AKI in ICU Chinese population	159 AKI patients (88 female and 71 male) admitted in ICU compared to 120 age-matched healthy volunteers (50 female and 70 male)	159 patients	SP-D polymorphism Thr11Met and Thr160Ala	SP-D-Thr11Met genotype was more susceptible to AKI	KDIGO
**Fatani, et al.,** **2018**	Prospective Study	Critical ill patients Severe sepsis induced AKI	200 critically-ill patients (112 had severe sepsis and septic shock and 88 were septic)	127 patients	TNF-α rs 361525	TNF-α rs 361525 was significantly associated with AKI	RIFLE
**He, et al.,** **2018**	Retrospective Case/Control Study	Children with AKI	1138 children with AKI and 1382 non-AKI controls.	1138 Children	TNF-α, IL6, IL10, IL18, NFKB1 and NFKBIA	NFKB1 rs28362491, NFKBIA rs2233406 and NFKBIA rs696 were associated with AKI in Children	pRIFLE
**Aghakhani Chegeni, et al.,** **2018**	Prospective Cohort Study	Iraninan patients undergoing Cardiac surgery	123 Patients undergoing CABG	63 Patients	STAT3 polymorphism	Rs1053004 GG genotype significantly decreased CSA-AKI risk	AKIN
**Westphal, et al.,** **2019**	Prospective, double-blind, multicenter, randomized trial (RIPHeart) GWAS.	Myocardial infarction, atrial fibrillation, acute stroke, acute kidney injury and delirium after cardiac surgery	1170 patients of both genders (871 males, 299 females) undergoing elective cardiac surgery	52 Patients	547,644 variants	PHLPP2, BBS9, RyR2, DUSP4 and HSPA8, associated with new onset of atrial fibrillation, delirium, myocardial infarction, AKI and stroke after cardiac surgery.	---
**Vilander, et al.,** **2019**	Prospective, observational Finnish Acute Kidney Injury (FINNAKI) study	Cohort of Finnish critically ill patients	2647 Critical ill patients without chronic kidney disease	625 patients	TNF-α (rs1800629), IL6 (rs1800796, rs1800795, rs10499563, rs1474347, rs13306435, rs2069842 and rs2069830), IL-8 (rs4073), IL10 (rs1800896), NOS3 (rs2070744), NFKB1A (rs1050851), AGT (rs699 and rs2493133), VEGFA (rs2010963 and rs3025039), EPO (rs1617640), SUFU (rs10748825), HIF1-α (rs11549465), PNMT (rs876493), MPO (rs7208693), COMT (rs4680), HSPB1 (rs2868371), SP-D (rs2243639 and rs721917), HAMP (rs10421768) and BBS9 (rs10262995) genes	rs1800629 in TNF-α; and rs1800896 in *IL-10 were not associated to* AKI	KDIGO
**Averdunk, et al.,** **2020**	Prospective, double-blind, multicenter, randomized trial RIPHeart Study	CSA-AKI	1116 patients undergoing cardiac surgery	170 Patients	MIF CATT5–7 (rs5844572/rs3063368, “-794”) and G > C single-nucleotide polymorphism (rs755622,-173)	The MIF CATT7 allele associates with a higher risk of AKI and death after cardiac surgery	KDIGO

Abbreviations: ACE, Angiotensin Converting Enzyme gene; AGT, Angiotensinogen gene; AGTR1, Angiotensin II Receptor, Type 1 gene; AKI, acute kidney injury; AKIN, Acute Kidney Injury Network; APOE, apolipoprotein E gene; AASK, American Study of Kidney Disease and Hypertension; BCL-2, B-cell CLL/lymphoma 2 gene; BBS9, Bardet-Biedl Syndrome 9 gene; CABG, coronary artery bypass graft; CAT, Catalase gene; CKD, Chronic kidney disease; CRIC, Chronic Renal Insufficiency Cohort; CSA-AKI, Cardiac surgery associated-AKI; COMT, Catechol-O-methyltransferase gene; Cr, creatinine, CYBA, Cytochrome b245, α subunit gene; DUSP4, Dual Specificity Phosphatase 4 gene; eGFR, estimated Glomerular Filtration Rate; eNOS, endothelial Nitric Oxide Synthase gene; EPO, Erythropoietin gene; ESRD, end-stage renal disease; GPX, glutathione peroxidase gene; GWAS, Genoma Wide Association Study; HAMP, Hepcidin antimicrobial peptide gene; HIF1- α, Hypoxia-Inducible Factor 1- α gene; HLA-DRB1, Human Leukocyte Antigen –Major Histocompatibility Complex, Class II, DR beta 1 gene; HSPA8, Heat Shock Protein 8 gene; HSPB1, Heat Shock Protein family B (small) member gene; IFN-γ,Interferon-γ gene; IL-6, Interleukin-6 gene; IL-18, Interleukin-18 gene; IL-10, Interleukin-10 gene; ICU, Intensive care unit; IRF2, Interferon Regulatory Factor 2; KDIGO, Kidney Disease Improving Global Outcome; KLK1, Kallikrein 1 gene; LTA, Lymphotoxin α gene; MDRD, modification of diet in renal disease; MIF, Macrophage Migration Inhibitory Factor gene; MnSOD, Manganese superoxide dismutase; MPO, Myeloperoxidase gene; MYH9, Muscle myosin heavy chain 9 gene; NADPH, Nicotinamide Adenosine Dinucleotide Phosphate gene; NFκB, Nuclear Factor-κB; NOS3, Nitric Oxide Synthase 3 gene; PBEF, Pre-B cell colony-enhancing factor gene; PHLPP2, PH domain and leucine rich repeat protein phosphatase 2 gene; PNMT, Phenylethanolamine N-methyltransferase gene; pRIFLE, Pediatric RIFLE; PTCH1, Patched homolog 1 gene; PTCH2, Patched homolog 2 gene; SERPINA4, Serpin Peptidase Inhibitor, Clade A (α-1 antiproteinase, antitrypsin) Member 4 gene; SERPINA5, Serpin Peptidase Inhibitor, Clade A (α-1 antiproteinase, antitrypsin) Member 5 gene; RIFLE, Risk Injury Failure, Loss, End-Stage; RIPHeart, Remote Ischemic Preconditioning Heart Study; RRT, Renal replacement therapy; RYR2, Ryanodine receptor 2 gene; SIK3, Salt-Inducible Kinase 3 gene; SMO, Smoothened gene; SNP, Single nucleotide polymorphism; SP-D, Surfactant protein-D gene; STAT3, Transducer and activator of Transcription 3 gene; SUFU, Suppressor of Fused homolog gene; TBX1, T-Box Transcription Factor 1 gene; TGF-β, Transient Growth Factor-β gene; TNF-α, Tumor Necrosis Factor α gene; US, United States; VEGF, Vascular Endothelial Growth Factor gene.

## Data Availability

Not applicable.

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
