# Peer review of "Genetic Susceptibility to Acute Kidney Injury"

_jcm, 2021, doi:10.3390/jcm10143039_

Round 1
Reviewer 1 Report
I read with interest the overview by Christian Ortega-Loubon and colleagues.
The Authors did an interesting analysis about the genetic polymorphisms potentially related to acute kidney injury
The overview is well written and complete
I suggest only minor comments:
- please, define acute kidney injury and the main definition applied in the studies that you analyzed
- please, try to better expalin how genetic determinants may interact with environmental determinants
- please, try to explain how the knowledge of these genetic determinants may influence the management of patients
- some Authors did not consider AKI mainly determined by contrast media administration, but by baseline patient's characteristics (i.e. age, baseline renal function, type of exam requiring contrast media, clinical condition, etc.). Your data suggests the presence of factors that interacting with contrast media induces AKI. Please, discuss it
Author Response
Dear Reviewer,
Thank you for your recommendations. We greatly appreciate it.
- We have included a table with all the studies detailing the AKI definition used in every clinical setting.
- The interaction between the genetic and the environment has been included in the introduction.
- How the management of this patient can be changed due to a better understanding of these genetic determinants is included in the conclusion of the manuscript.
- Thank you for your recommendation. We have included the contras induced-AKI discussion as suggested.
Reviewer 2 Report
The authors thoroughly list and describe studies which have investigated genetic association with risk of AKI. They also list the known functions associated with the genes.
However, the whole manuscript rather reads as a listing, but given that the manuscript is supposed to be a review, a comprehensive discussion is missing.
The strengths and limitations of the GWAS with the greatest impact should be discussed. Effect sizes should be discussed. Additionally, authors should mention the setting in which AKI has been investigated in each specific study: AKI after surgery (which surgery), AKI in sepsis/infection? Was there also an association with more severe AKI requiring dialysis?
I also miss a comparative table clearly showing all genes/proteins and the related evidence (including type of study, cohort size, effect sizes, clinical setting of the study).
How about polymorphisms in genes coding for proteins that have recently been suggested as biomarker of AKI (such as NGAL, KIM-1, TIMP-2 and IGFBP2).
We have studied polymorphisms in the MIF (macrophage migration inhibitory factor) promotor as a genetic susceptibility gene of AKI. Why have authors not included MIF?
I also miss a summary the most important genes/markers and the conclusion that can be drawn from that.
Author Response
Thank you for your time to review the manuscript. We greatly appreciate it.
Reviewer 2 reply
The authors thoroughly list and describe studies which have investigated genetic association with risk of AKI. They also list the known functions associated with the genes.
However, the whole manuscript rather reads as a listing, but given that the manuscript is supposed to be a review, a comprehensive discussion is missing.
Response: We have changed the whole article structure, so that it is no more a listing article but a comprehensive review discussion.
The strengths and limitations of the GWAS with the greatest impact should be discussed. Effect sizes should be discussed. Additionally, authors should mention the setting in which AKI has been investigated in each specific study: AKI after surgery (which surgery), AKI in sepsis/infection? Was there also an association with more severe AKI requiring dialysis?
Response:
The limitations of GWAS is included in the introduction of the article.
The clinical setting fo the different scenarios in each study is included in the table suggested to add by the reviewer.
I also miss a comparative table clearly showing all genes/proteins and the related evidence (including type of study, cohort size, effect sizes, clinical setting of the study).
Response: The comparative table was included as suggested.
How about polymorphisms in genes coding for proteins that have recently been suggested as biomarker of AKI (such as NGAL, KIM-1, TIMP-2 and IGFBP2).
Response: Polymorphisms for the newly biomarkers was not included because we didn't find any specific variants identified in the literature.
We have studied polymorphisms in the MIF (macrophage migration inhibitory factor) promotor as a genetic susceptibility gene of AKI. Why have authors not included MIF?
Response: MIF polymorphism was included in the Review Article. Thank you for the suggestion.
I also miss a summary the most important genes/markers and the conclusion that can be drawn from that.
Response: The most important genes were included in the conclusion, as well as the take to home idea.